# Process evaluation for the STAMINA randomised controlled trial: A protocol

Saïd Ibeggazene[1]*, Sophie Reale[1], Alison Scope[1], Grace Price[1], Eileen Sutton[2], Liam Bourke[1], Michelle Collinson[3], Jamie Stokes[3], Liz Steed[4], Derek Rosario[5], Amanda J. Farrin[3], Steph J.C. Taylor[4]

1 Department of Allied Health Professions, College of Health Wellbeing and Life Sciences, Sheffield Hallam University, Sheffield, United Kingdom, 2 Bristol Population Health Sciences, University of Bristol, Bristol, United Kingdom, 3 Clinical Trials Research Unit, Leeds Institute of Clinical Trials Research, University of Leeds, Leeds, United Kingdom, 4 Wolfson institute of Population Health, Queen Mary University of London, London, United Kingdom, 5 Department of Urology, Royal Hallamshire Hospital, Sheffield Teaching Hospitals NHS Foundation Trust, Sheffield, United Kingdom

* s.ibeggazene@shu.ac.uk

## Abstract

### Background

STAMINA is a randomised controlled trial of a complex lifestyle intervention incorporating exercise prescription into a prostate cancer care pathway. The 12-month intervention aims to improve disease specific quality of life and reduce fatigue of people receiving androgen deprivation therapy for prostate cancer. Previously published work outlines the development of the trial intervention which included recruitment and training of healthcare professionals and exercise professionals to embed a lifestyle intervention and referral pathway within NHS prostate cancer care.

### Methods

A mixed-methods process evaluation, embedded within the STAMINA trial, will be conducted to assess quantitative process outcomes (recruitment, intervention reach, dose and fidelity), together with up to 45 qualitative interviews with patients, healthcare professionals and exercise professionals. Interviews will explore the perceptions and experiences of those involved in the STAMINA trial, and the organisational implications of embedding and sustaining the intervention. Quantitative process data will be analysed descriptively. Qualitative interview data will be analysed before trial outcomes are known using an inductive and deductive approach. Findings from the different elements will be reported separately and then integrated to inform interpretation of trial outcomes.

**Data availability statement:** No datasets were generated or analysed during the current study. All relevant data from this study will be made available upon study completion.

**Funding:** This work is funded by the NIHR Programme Grants for Applied Research (PGfAR) RP-PG-1016-20007. The views expressed are those of the author(s) and not necessarily those of the NHS, the NIHR or the Department of Health and Social Care. The funders had no role in study design, data collection and analysis, decision to publish, or preparation of the manuscript.

**Competing interests:** The authors declare one competing interest. Professor Liam Bourke is funded by the NIHR and acts as a scientific consultant for Boston Scientific Corp.

## Conclusion

This process evaluation protocol provides a detailed description of relevant data collection methods and trial processes of the STAMINA randomised controlled trial which will allow us to determine whether the intervention can be delivered with fidelity, is acceptable to patients, healthcare professionals and exercise professionals, and understand the implications for embedding and sustaining the intervention in the routine care.

## Trial registration

ISRCTN 46385239, registered on 30/07/2020. Cancer Research UK 17002, retrospectively registered on 24/08/2022.

## Background

Prostate cancer is a long-term condition which is estimated to affect more than 1.4 million people globally [1], a figure which is projected to double by 2040 [2]. In the UK, like many developed countries, prostate cancer is the most common male cancer and its prevalence is projected to increase until 2035 at least, whilst prostate cancer-related mortality rates continue to decline [3]. Consequently, the prostate cancer survivorship population is likely to continue to expand. Though the impact of prostate cancer on a person's life varies substantially, advances in prostate cancer treatment have reduced mortality, meaning much of the impact of living with prostate cancer is iatrogenic in nature.

A key effective treatment of advanced prostate cancer is androgen deprivation therapy (ADT) which can be administered for up to two decades in combination with shorter courses of other treatments including radiotherapy, chemotherapy, and androgen receptor inhibitors. Notwithstanding the negative sequelae of other treatments, ADT is associated with increased burden of adverse effects including: fatigue, fat mass gain, loss of lean mass, hot flushes, bone fracture risk, diabetes, cognitive dysfunction, depression and sexual dysfunction [4]. Individuals may experience one or many of these effects leading to negative impacts on quality of life. At present, the only evidence-based intervention to improve quality of life and fatigue in this population is supported exercise training [5]. This has led to 12 weeks of supported aerobic and resistance exercise training being recommended for people receiving ADT in national and international guidelines [6,7]. Despite this, access to such a service in the UK is extremely scarce [8].

To address this need, we undertook a programme of work with the aims of defining, developing, implementing, and evaluating a complex lifestyle intervention incorporating exercise prescription into a care pathway for people receiving ADT for prostate cancer. We intended for the intervention to be embedded within NHS prostate cancer care pathways and implementable across the UK. The culmination of this work is the STAMINA randomised controlled trial (RCT). This paper describes the protocol for the parallel process evaluation.

## Trial Status

The RCT protocol has been published [9] (ISRCTN 46385239, registered on 30/07/2020. Cancer Research UK 17002, retrospectively registered on 24/08/2022). 700 participants were recruited between January 2022 and June 2023 and the trial is now in follow-up. Results of the clinical effectiveness and economic evaluations of the STAMINA intervention will be made available once trial follow-up is complete.

## The STAMINA randomised controlled trial

The STAMINA RCT is investigating the clinical and cost effectiveness of a 12-month lifestyle intervention for men receiving ADT for prostate cancer. We hypothesise that the intervention will improve disease-specific quality of life and/or reduce fatigue after 12 months (FACT-Prostate [10] & FACIT-Fatigue [11] questionnaires, respectively). The trial will also assess whether the intervention has an impact on blood pressure, body mass, physical function, abdominal adiposity, self-reported physical activity, behavioural determinants of exercise behaviour, fear of cancer recurrence and the burden of side effects from ADT.

NHS healthcare professionals at participating sites were invited to receive training to enhance their knowledge, skills, and confidence regarding endorsing exercise to this population during usual care in line with NICE guidance prior to participant recruitment. All participants received a standardised purpose-made information booklet with information about exercise and dietary recommendations for this population plus related booklets from UK cancer charities [12–15]. Accordingly, the control is described as optimised usual care (OUC). Trial participants are randomised to receive OUC or the STAMINA lifestyle intervention (SLI+OUC).

## The STAMINA lifestyle intervention

The STAMINA lifestyle intervention (SLI) is a complex intervention facilitated by NHS healthcare professionals (HCPs) and exercise professionals (EPs) from Nuffield Health fitness centres. The development and refinement of the complex intervention components have been published previously [16–18]. In brief, intervention development was underpinned by the Theoretical Domains Framework [19] and guided by the Behaviour Change Wheel [20,21]. Complimentary stakeholder workshops were delivered to ensure future implementation was considered at all stages of development and refined based on Normalisation Process Theory (NPT) [22].

Both HCPs and EPs receive bespoke behaviourally informed training and ongoing support to enable them to jointly (but separately) implement the intervention. In this way, there are three components to the complex intervention that coalesce to produce the intended changes in participant outcomes.

We have developed a logic model (Fig 1) which outlines the assumed causal chain of events in SLI.

## Healthcare professional training

Healthcare professionals from NHS Urology and Oncology departments (including clinical nurse specialists, consultants, radiographers, and prostate cancer support workers) were recruited and trained to refer NHS patients who were receiving ADT for prostate cancer to the trial and to endorse and support patients to exercise. Trained HCPs have the potential to influence participants exercise behaviours prior to randomisation, this component of the intervention was deemed necessary for the successful implementation of SLI. Ongoing support for participants to exercise is provided as routine clinical appointments.

A behaviourally informed training package underpinned by the Theoretical Domains Framework [19] was developed for HCPs to support the delivery of NICE recommendations including identifying eligible patients suitable for exercise, recommending exercise, providing information, making an exercise referral and providing support [16]. The training package was delivered at site or remotely via video-conference in an up to three-hour interactive training package. NHS staff received ongoing support from the intervention training team.

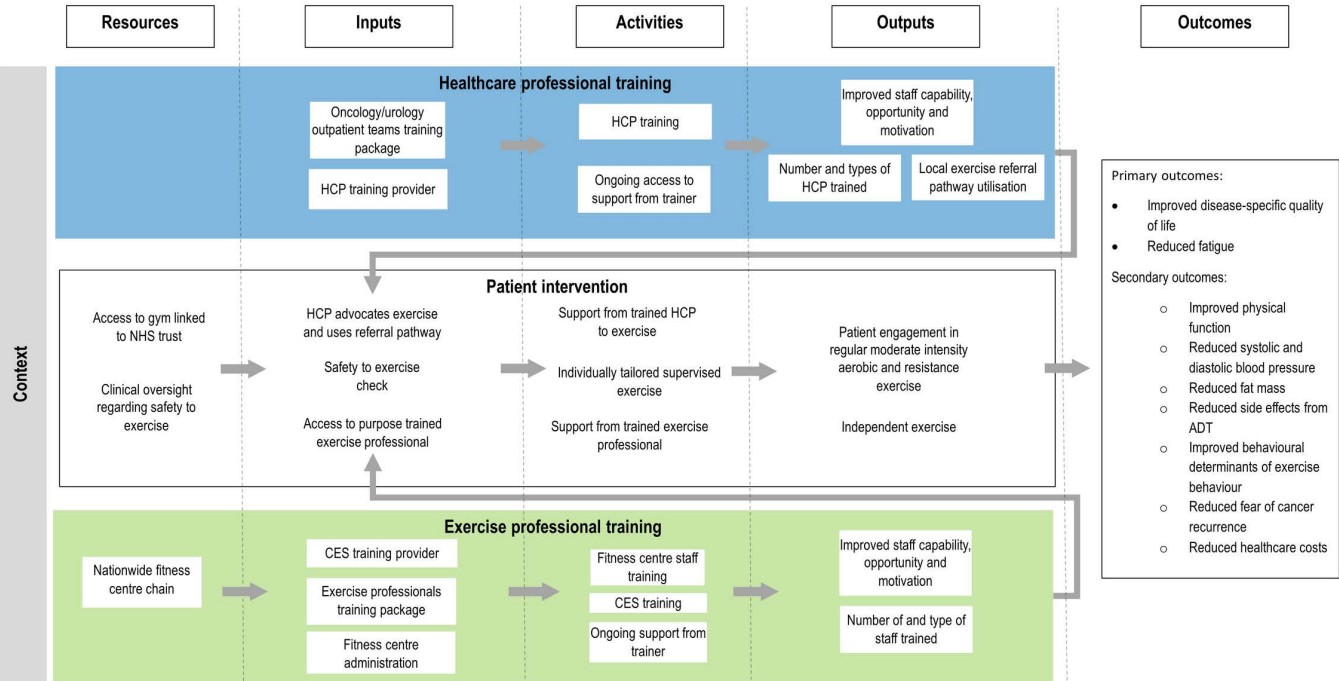

**Fig 1. STAMINA programme logic model.**

## Exercise professional training

A range of staff (including front-of-house staff, management, and the fitness team) at Nuffield Health sites were invited to complete three short online training modules providing introductory content regarding who the intervention was designed for and how to operationalise the intervention locally. Subsequently, the fitness team were invited to a full day interactive training session to develop their skills, knowledge, and confidence to deliver SLI. Upon satisfactory completion of training, EPs were designated the title of Clinical Exercise Specialist (CES). The CES role included providing an induction to the gym, delivering supervised exercise and behavioural support to trial participants and recording data in line with the trial protocol. Nuffield Health staff received ongoing support from the intervention training team.

## The participant STAMINA lifestyle intervention

The published trial protocol provides a full description of the intervention [9].Participants randomised to SLI are provided with an induction to the gym and facilities, 12 weeks of twice weekly scheduled individually tailored supervised exercise, a review of progress with behavioural support at weeks 2, 4, 6 and 12 and every three months thereafter and 3–9 scheduled supervised exercise sessions over the remaining nine months of the programme (during which time the participant is encouraged to independently exercise twice weekly). The first 12 "weeks" of the programme can be completed over a 14-week period to allow for a run-in period whereby the intensity/volume/frequency of sessions may begin below the initial recommended prescription and gradually increase to the full prescription. The degree of reduction of the prescription and the rate of increase in prescription parameters is at the discretion of the CES responsible for the participant. Sessions in this period are initially delivered on a one-to-one basis but are progressed to group exercise when the CES deems the participant to require less intensive supervision. Modification of the programme will be considered for individual cases (e.g. to provide remote sessions). Each participant may receive supervision from multiple members of the CES team

during the intervention. Exercise sessions generally last 60 minutes and include 30–45 minutes of aerobic exercise and up to four sets of resistance exercises for major muscle groups.

## Process evaluation

This protocol outlines an embedded parallel process evaluation to the STAMINA trial which was designed following initial intervention development work and a pre-pilot study [16–18].

### Aims and objectives

The aims of this process evaluation are:

- To understand trial recruitment performance.

- To describe intervention reach, dose delivered, and dose received.

- To describe the fidelity of intervention delivery.

- To understand how the intervention was experienced and understood by patients using semi-structured interviews (underpinned by the Theoretical Framework of Acceptability).

- To explore the organisational implications for embedding and sustaining the intervention in preparation for wider NHS roll-out using semi-structured interviews (underpinned by consideration of Normalization Process Theory).

## Methods

Informed by MRC guidance on evaluation of complex interventions and process evaluation and the Linnan and Steckler process evaluation framework [23–25], we are undertaking a mixed-methods process evaluation. All process evaluation data will be collected prospectively and without knowledge of trial outcomes, either by participants or evaluators. Ethics approval was granted West of Scotland Research Ethics Committee (1 20/WS/0069) and written, informed consent to participate will be obtained from all participants. Members of the process evaluation team are involved in the delivery of the trial. SI, SR, AS and GP were involved in contacting and consenting (SI & GP) prospective trial participants. SR is responsible for providing training and intervention support to EPs and conducting interviews with HCPs and trial participants. AS is responsible for providing training and intervention support to HCPs and conducting interviews with EPs and trial participants. SI is responsible for delivering safety to exercise checks and defining tailored exercise parameters for participants randomised to SLI.

A summary of the process evaluation components which will be targeted for data collection have been mapped against the key components of process evaluation according to Linnan and Steckler [25] (Table 1). Table 2 lists the data collection methods used, with each method cross-referenced to the process evaluation components which they address in Table 1. The timescales for these process evaluation methods have been mapped onto the main trial timescales and detailed in Appendix 1.

### Qualitative data descriptions and analysis

**Interview method.** We will conduct one to one interviews (telephone or face to face) or focus groups (depending on interviewee preference and feasibility) with a purposive sample of *up to*: 15 SLI participants, 10 OUC participants; 10 CESs and 10 HCPs. Interviews and focus groups will be conducted at different times across the trial to help capture any temporal effects (including referral and reporting processes). Topic guides have been developed in accordance with the NPT (for professional interviews) and Theoretical Framework of Acceptability (TFA) [26] (for trial participant interviews) to explore implementation and acceptability of the intervention respectively. The same questions will be asked to SLI

Table 1. Process evaluation components.

| MRC/ Linnan & Steckler component | Participant intervention | Healthcare professional training | Exercise professional training |
|---|---|---|---|
| Recruitment | • 1) Numbers of participants screened and their characteristics: treatments, type of ADT, other PCa treatments, age, ethnicity, referral pathways. | • 22) Number of potential NHS STAMINA sites and data related to site eligibility<br>• 23) Number of sites who agree to be part of STAMINA and approve recruitment to begin<br>• 24) Reasons given by sites for refusal to participate<br>• 25) Characteristics of NHS sites recruited e.g. size, patient capacity, hospital type, rural vs urban, NH partner site nearby etc.<br>• 26) Number of sites willing to take part vs. number of sites recruited<br>• 27)NHS primer meeting attendance (invited vs. attended) | • 40) Number of NH sites made available to STAMINA trial by NH and number partnered with NHS site<br>• 41) Characteristics of NH sites involved in STAMINA (e.g. location in relation to NHS site, size of fitness team (no. of EPs on site), are sites delivering on other internal NH 'flagship programmes' etc.<br>• 42) NH primer meeting attendance (invited vs. attended)<br>• 43) Number of NH EPs in NH sites |
| Reach | • 2) Number/proportion of men referred to study team<br>• 3) Number of men on ADT interested in participating in STAMINA<br>• 4) Reasons for ineligibility/ not interested in participating (if provided)<br>• 5) Number of participants eligible<br>• 6) Number of participants consented and randomised<br>• 7) Characteristics of randomised participants | • 28) Take up of training of HCPs in intervention sites, including characteristics and proportions of those invited vs. those attending (urology vs. oncology staff, consultant vs. nurse or other HCP etc). | • 44) Total number of NH staff attending training<br>• 45) Characteristics of EPs undergoing and passing training including length of time at NH, as EP, qualifications, previous experience with clinical populations, experience with flagship programmes and any remote delivery experience |
| Dose delivered | • 8) Number of supervised sessions and reviews offered by Nuffield Health staff | • 29) Number of training sessions delivered per site and per staff member<br>• 30) Descriptions of training delivery: when and in which sites training was delivered, training duration, mode of delivery, attendance, number of follow-up training sessions delivered etc.<br>• 31) Training facilitator characteristics<br>• 32) Follow-up intervention support provided | • 46) Number of training sessions delivered per site and per staff member<br>• 47) Descriptions of training delivery: when and for which sites training was delivered, training duration, mode of delivery, attendance, staff turnover, number of follow-up training sessions delivered, attendance etc.<br>• 48) Training facilitator characteristics<br>• 49) Follow-up intervention support provided |
| Dose received | • 9) Number of participants actually receiving the supervised intervention<br>• 10) Supervised exercise dose: Number of supervised training sessions attended, exercise intensity, aerobic exercise duration, resistance training volume<br>• 11) Intervention and delivery format (i.e. number and proportion of face-to-face vs remote sessions, number and proportion of group vs 1-2-1 sessions)<br>• 12) Details of (unsupervised) gym use and other physical activity by intervention participants<br>• 13) Behavioural support diary completion<br>• 14) Progress/summary report produced by CESs including number, timeliness, content and any evidence it has been acted on by HCPs | • 33) Assessment of change in behavioural determinants impacted/ targeted by the intervention<br>• 34) Engagement with follow-up support | • 50) Assessment of change in behavioural determinants impacted/ targeted by the intervention<br>• 51) Engagement with follow-up support |

(Continued)

**Table 1.** (Continued)

| MRC/ Linnan & Steckler component | Participant intervention | Healthcare professional training | Exercise professional training |
|---|---|---|---|
| Fidelity | • 15) Digital audio recordings of CES: patient induction and review sessions (from a sub-sample of N=5 sites).<br>• 16) Direct observations of CESs delivering supervised sessions (up to a maximum of 30 observations)<br>• 17) Video recording of remote intervention delivery sessions | • 35) Video recording of HCP training delivery | • 52) Video/audio recording of CES training delivery |
| Implementation | • 18) Adverse event reporting<br>• 19) Intervention dropout rates and reasons for dropout<br>• 20) Participants experience of ADT<br>• 21) Interviews exploring the acceptability of the STAMINA lifestyle intervention to participants | • 36) Recording the length of delay between receipt of training and first delivery of intervention<br>• 37) Interviews exploring perceptions and experiences of those involved in STAMINA, and the organisational implications of embedding and sustaining the intervention in preparation for a wider NHS roll-out of the programme.<br>• 38) Field notes following HCP training delivery<br>• 39) Field notes of support interactions with NHS teams | • 53) Recording the length of delay between receipt of training and first delivery of intervention<br>• 54) Number of participants seen by each CES<br>• 55) Format and amount of the participant intervention delivered by each CES<br>• 56) Length of time CESs remain in study delivering intervention<br>• 57) Reasons for CES discontinuation in trial (if applicable)<br>• 58) Interviews exploring the perceptions and experiences of those involved in STAMINA, and the organisational implications of embedding and sustaining the intervention in preparation for a wider NHS roll-out of the programme.<br>• 59) Field notes |

ADT– Androgen Deprivation Therapy, CES – Clinical Exercise Specialist, EP – Exercise Professional, HCP – Healthcare professional, NH – Nuffield Health, NHS – National Health Service, PCa – Prostate Cancer.

**Table 2. Process evaluation data collection methods.**

| Data collection methodology | Target population | Data collection methods (components) |
|---|---|---|
| Qualitative analysis | Participants | • Behavioural support diary completion (13)<br>• Participant interviews (21) |
| | Exercise professional team | • Interviews (58)<br>• Video/audio recordings of intervention training (52)<br>• Direct observations of CESs delivering supervised sessions (16)<br>• Audio recordings of CESs delivering review sessions (15)<br>• Remote supervised exercise session recordings (17)<br>• Intervention support log (49, 51)<br>• Field notes (59) |
| | Healthcare professional team | • Interviews (37)<br>• Video recordings of intervention training (35)<br>• Intervention support log (32, 34)<br>• Field notes (38–39) |
| Quantitative analysis | Participant | • Screening logs at site (1, 2)<br>• Referrals processing databases and CRFs (2–6,36)<br>• Baseline participant characteristics CRF (7)<br>• Participant withdrawal CRF (19)<br>• Adverse Event reporting (18)<br>• ADT symptom index (20) – S1 File<br>• Godin Leisure-time exercise questionnaire (12)<br>• Participant intervention logbooks (8–11, 53–56)<br>• Swipe card data (logs of swipes of a membership card to access the gym) (12)<br>• Progress/summary reports (14) |
| | Exercise professional team | • Nuffield Health site screening log (40)<br>• Nuffield Health site characteristics CRF (41)<br>• Training facilitator characteristics CRF (48)<br>• Meeting and Training attendance logs (42–44, 46, 47,51)<br>• EP personal details CRF (45)<br>• EP Theoretical Domains Framework questionnaires (50)<br>• Authorised personnel logs (56–57) |
| | Healthcare professional team | • Site feasibility questionnaires (22–26)<br>• Training facilitator characteristics CRF (31)<br>• Meeting and Training attendance logs (28–30)<br>• HCP personal details CRF (28)<br>• HCP Theoretical Domains Framework questionnaires (33) |

ADT – Androgen Deprivation Therapy, CES – Clinical Exercise Specialist, CRF – Case Report Form, EP – Exercise Professional, HCP – Healthcare professional, TDF – Theoretical Domains Framework.

and OUC participants except for questions specific to the intervention which will be omitted for those in the OUC arm. All participants must provide consent and remain in the trial before interview/focus group. Copies of the topic guides can be found in the Supplementary materials.

Interviews will be audio recorded, following agreement from the interviewee, using an encrypted dictaphone and will be professionally transcribed. During transcription, any potentially identifying information that may be contained in the interview discussions will be anonymised or removed. Only the research team and the transcriber will listen to the interview audio files and have access to the transcripts. All audio files, transcripts and field notes will be securely transferred and stored in encrypted format, accessible only to members of the study team requiring such access.

**Interview sampling.** Participants will be purposively sampled based on their age, geographical location, and stage of the intervention/duration of follow-up. SLI participants will also be purposively sampled based on their adherence to the intervention (e.g., low, moderate, or high).

CESs will be eligible for interview once they have completed intervention training and delivered at least 12 weeks of supervised exercise to five or more participants. CESs will be purposively sampled based on details collected during staff training (i.e., role, geographical location, tier of club, gender, experience).

HCPs will be eligible for interview once they have completed intervention training. HCPs will also be purposively sampled based on details collected as part of staff training (i.e., role, speciality, geographical location, hospital type, experience).

**Interview analysis.** We will take an inductive and deductive approach to thematic analysis guided by Braun and Clarke's six phases [27]. NPT constructs and sub-constructs (Table 3) will provide the framework for analysis of professional interviews whilst the TFA (Table 4) will provide the framework for participant interviews [26]. We will also consider data that falls outside of the priori codes to derive themes and concepts. At least 20% of data will be independently double or triple coded. Decisions, disagreements, and development of the frameworks will be documented in a coding manual at every phase of analysis. Once the transcripts have been coded, for each of the three participant groups, all data will be collated into themes either prespecified by the NPT and the TFA or falling outside of these frameworks. There will then be a process of checking to ensure themes work in relation to coded extracts and the entire data set. A thematic map will be generated, followed by ongoing analysis to refine the specifics for each theme [27].

**Assessment of fidelity.** Fidelity will be conceptualised in accordance with the National Institute of Health Behavioural Change Consortium framework [28]. This will explore whether the intervention was delivered as intended and the

**Table 3. Understanding intervention embedding in NHS practice informed by Normalization Process Theory.**

| Component | Example |
|---|---|
| Coherence | Understanding the purpose, value, and benefits of the STAMINA lifestyle intervention, including roles and responsibilities |
| Cognitive participation | Initiating and sustaining buy-in, STAMINA 'champions' at NHS sites, individual engagement with STAMINA |
| Collective action | How STAMINA works in day-to-day practice (including roles/resources), communication pathways, work people must do to make STAMINA happen |
| Reflexive monitoring | Appraisal of the benefits and costs of STAMINA, including refinements recommended for a future roll-out |

**Table 4. Theoretical Framework of Acceptability constructs and example questions.**

| Theoretical framework of acceptability construct | Example |
|---|---|
| Affective attitude | How participants feel about SLI and OUC, including change in feelings over time |
| Burden | The perceived amount of effort required to participate in SLI and OUC, including research related processes |
| Ethicality | How does SLI/OUC align with the participants beliefs and values, including what is important to them |
| Perceived effectiveness | Actual or perceived benefits/outcomes of SLI and OUC, including change over time and any uncertainty related to outcomes |
| Intervention coherence | The extent to which participants understand the trial, and components of SLI/OUC, including information provided by HCPs and EPs |
| Self-efficacy | The participants confidence to perform the required behaviours of SLI and OUC, including confidence to maintain behavioural change |
| Opportunity costs | Anything that participants have missed/given up to participate in SLI or OUC and any reason for non-attendance/ participation |
| Affective attitude | How participants feel about SLI and OUC, including change in feelings over time |
| Burden | The perceived amount of effort required to participate in SLI and OUC, including research related processes |

proportion of target behaviours implemented. In multi-site studies, it is particularly important to assess treatment fidelity because if treatment effects vary by site, we can observe whether this is related to variations in the delivery of treatment. For the STAMINA process evaluation, analysis will capture data at three levels: training delivery, training receipt and intervention delivery.

**Fidelity: Training delivery.** HCP and EP training will be (digitally or audio) recorded to capture information about how professionals were trained and whether training was standardised across sites. The focus will be on the training facilitators as opposed to the professionals in attendance. Two coding checklists will be developed to assess what was delivered in training (content and behaviour change techniques) compared to what was intended to be delivered as defined by the HCP and EP training manual. Details about facilitator characteristics and how the session was delivered will be captured on paper forms.

**Fidelity: Training receipt.** Assessment of cognitive, affective, social and environmental influences of HCP and EP behaviour will be captured in a questionnaire developed based on the TDF [29] (Appendix 2–4). Questionnaires will be distributed immediately prior to training and repeated immediately following training. The questionnaires will be repeated at 3-, 6- and 12-months post training, for EPs, to monitor influences of behaviour over time. We will also record the length of delay between receipt of training and first delivery of intervention.

**Fidelity: Intervention delivery.** The gold standard method for capturing fidelity to intervention delivery is coding of digital recordings, evaluated according to criteria developed *a priori* [30]. We will purposively sample five Nuffield Health fitness and wellbeing clubs (based on their tier, geographical location, facilities, staffing, and experience delivering clinical programmes) and, via an encrypted dictaphone, obtain audio recordings of one-to-one sessions (i.e., inductions and progress reviews) between CESs and SLI participants who have consented to recordings. This method is not feasible for the assessment of supervised exercise sessions that are delivered on a busy gym floor with members of the public in attendance. Therefore, fidelity of supervised exercise sessions will be assessed using pre-specified checklists completed during session observations (either via in person live observation or from a digital session recording) (up to 30 in total) by SR and AS. Checklists will be developed in line with guidance [28] to score adherence and quality of delivery of target behaviours and behaviour change techniques as per protocol. We will also record instances where additional/ other behaviours or behaviour change techniques are delivered to allow for evaluation of treatment differentiation. Contemporaneous field notes regarding the context and other related information will be recorded and stored on a secure drive.

**Fidelity: Analysis.** Once developed, checklists will be used to score each core behaviour and behaviour change technique between 0 and 2 (0 = not delivered but applicable, 1 = limited delivery, 2 = full/appropriate delivery). If the behaviour is deemed not applicable, then NA will be written. Scores will be summed and converted to a percentage. Levels of treatment fidelity will be interpreted in line with the literature as: 80–100% adherence interpreted as 'high' fidelity, 51–79% as 'moderate' and 0–50% as 'low' fidelity [28,31]. This process will be completed for all recordings of professional training delivery and session observations however for the audio recordings of CES consultations we will analyse a purposive sub-sample [32,33]. The purposive sample will include a minimum data set to reflect 10% of participants receiving SLI and 10% of CESs delivering the intervention and sampling of all time-period interactions. A full sampling framework is detailed in Appendix 5. At least 10% of recordings will be independently coded by two reviewers. A coding manual will be created and updated during piloting and beyond ensuring all key decisions are documented during the analysis.

## Quantitative data descriptions and analysis

Analyses of quantitative data will be descriptive in nature. Participant recruitment (rates and characteristics) was monitored via screening logs at site and using routine data (referral outcomes, eligibility, reasons for not taking part, recruitment timelines) from a referrals management system used by the research team. Demographic and clinical characteristics

for randomised participants were collected at baseline by trial researchers. Participant withdrawals from the trial or the intervention (including reasons for withdrawal) and related unexpected serious adverse events are recorded by researchers on an ongoing basis during the trial.

Dose monitoring data is drawn from several sources including: participant intervention logbooks, session attendance reports (online form), swipe card data (logs of swipes of a membership card to access the gym) and progress/summary reports produced by CESs including number of sessions prescribed versus attended, timeliness, content and any evidence it has been acted on by HCPs. These data will also allow monitoring of the delivery of the intervention by CESs. Adherence to SLI will be calculated as the number of supervised SLI sessions participants attended as a proportion of those scheduled.

The TDF questionnaires for HCPs (Appendix 3) and EPs (Appendices 2 & 4) will be scored by summing the scores of questions within each domain and dividing by the number of questions within that domain [34]. TDF scores will be reported for questionnaires completed immediately pre and post training and again at 3-, 6- and 12-months post training.

To characterize the NHS sites involved in the trial, prospective sites completed a site feasibility questionnaire, and further characteristics were gathered from publicly available data [35]. Characteristics of the Nuffield health sites were also collected via a case report form which was completed by fitness managers or general managers at site during set-up.

To better understand the implementation of the intervention and training of the two groups of intervention facilitators, data was collected about the delivery of training (training attendance logs, training facilitator characteristics forms) and characteristics of the staff being trained (HCP personal details form, EP personal details form). The length of time CESs remained in the trial delivering the intervention and the reasons for CES discontinuation were recorded where available.

## Data synthesis

The mixed-method synthesis will seek to address the aims of this process evaluation, providing explanations for results and recommendations for future practice. The main trial findings will be analysed independently of the process evaluation findings. We aim to produce a high-quality, integrated evaluation of the trial and intervention processes informed by a clear conceptual framework providing evidence to support our interpretations and providing context. Where possible, data from multiple sources will be combined to triangulate our understanding of the process evaluation components (See Table 1). We will use integrated analysis to produce a narrative synthesis drawing upon learning from all data sources to refine the analysis and inform a critical interpretation for the initial programme theory [36]. For example, where appropriate, the qualitative analysis may help us select variables to explore quantitative associations between factors that influence the delivery of the intervention and a participant's engagement with it. Similarly, descriptive quantitative data may provide important context for qualitative observations and allow for a more informed interpretation.

## Limitations

Our process evaluation approach is not without limitations. Context is one process evaluation component from the Linnan and Steckler/MRC framework for which we have not specified the method of formally capturing data, however we anticipate that this will be informed by data from multiple existing sources (e.g., interviews and observations). One limitation of our approach to fidelity concerns the absence of assessment of components of the intervention that are facilitated by HCPs during interactions with participants. In our feasibility study [18], we found collection of audio recordings of conversations between participants and trained HCPs to be difficult to implement in an unbiased manner due to resistance of clinical staff towards recording (and perceived monitoring) of normally confidential conversations and impracticalities around dictaphone use. Another limitation arises from a lack of resource for dedicated, independent process evaluation staff – however funding for such staff is limited by the cost envelope of the study and the inevitable re-distribution of resources following necessary adaptions to the trial design after the COVID pandemic. Many staff in the process evaluation team have been involved in intervention development and trial delivery which has the potential to create biases in the

collection and reporting of process evaluation data. In publishing this protocol, we intend to mitigate against the impact of these potential biases by acknowledging their potential in the data collection process for and by making explicit what data is to be collected, how each data type will be used to describe specific process evaluation components and how the process evaluation components and frameworks relate to each other. In addition, the intervention was developed prior to the COVID-19 pandemic, and it was not possible to re-pilot it in a post-pandemic context, thus potential for unanticipated consequences relating to the changes in attitudes towards engaging with the intervention and organisational changes within the NHS or Nuffield Health that took place in this period could not all be anticipated.

## Discussion

STAMINA is a complex lifestyle intervention that embeds an exercise prescription within NHS prostate cancer care pathways which is being evaluated in the STAMINA RCT [9]. To better understand the outcomes of the trial and the factors which may have influenced how, and how well, the intervention was delivered, we are conducting a process evaluation. This evaluation will systematically capture qualitative and quantitative data gathered from patients, healthcare professionals, healthcare organisations, exercise professionals and Nuffield Health our exercise delivery partner. We have defined the objectives of this process evaluation and have a clear conceptual framework from which to approach our analysis, however, the diverse and comprehensive choice of data collection methods used may provide us with additional opportunities to explore the "messy realities" that emerge throughout the course of the trial [37] which extend beyond these objectives. The design of this process evaluation will allow us to maximise the usefulness of the STAMINA trial by providing stakeholders and researchers with rich data and comprehensive analysis to understand its outcomes. Should the trial have a positive outcome this process evaluation will allow us the opportunity to provide informed recommendations for intervention implementation ahead of potential future roll-out. Through understanding the strengths and weaknesses of the intervention, how and to what extent it was delivered to and received by participants and how the intervention can be optimised to attain its intended purpose, we aim to provide a clinically and cost-effective exercise referral pathway for men receiving ADT for prostate cancer which improves quality of life and/or reduces fatigue.

## Supporting information

**S1 File. ADT symptom index.**
(DOCX)

**S2. Appendix 1–5.**
(DOCX)

**S3. STAMINA_Topic_Guide_Interviews_v5.0_20230705.**
(DOCX)

## Consent for publication

For the purpose of open access, the author has applied a Creative Commons Attribution (CC BY) licence to any Author Accepted Manuscript version of this paper, arising from this submission.

## Acknowledgments

We are very grateful for the substantial contributions made by many to the setting up of this trial and process evaluation: our Sponsor team at Sheffield Teaching Hospitals NHS Foundation Trust; our PSC chaired by Peter Sasieni, and other members Alison Birtle, Richard Bryant and Rachel Elliott; our PMG and STAMINA co-applicants (Patrick Doherty, Jenny Hewison, Janet Brown, David Meads, Diana Greenfield, Dylan Morrissey, Suzanne Hartley and Malcolm Mason); PPIE lay

 

member of our PSC (Geoff Ogden), and PMG/TMG (Tom Baker) who reports on behalf of the PPI Group, co-chaired with John Kidder and Chris Allen; colleagues at the University of Leeds CTRU and Sheffield Hallam University who supported development and implementation of the trial protocol. We are especially grateful to Nuffield Health for their tremendous support in the setting up and delivery of this trial. We would in particular like to thank Aidan Innes and Ben Kelly for their support, and all local Clinical Exercise Specialists and fitness managers.

## Author contributions

**Conceptualization:** Sophie Reale, Eileen Sutton, Liam Bourke, Michelle Collinson, Liz Steed, Derek Rosario, Amanda J Farrin, Steph JC Taylor.

**Funding acquisition:** Eileen Sutton, Liam Bourke, Michelle Collinson, Liz Steed, Derek Rosario, Amanda J Farrin, Steph JC Taylor.

**Investigation:** Sophie Reale, Eileen Sutton, Liz Steed.

**Methodology:** Saïd Ibeggazene, Sophie Reale, Eileen Sutton, Liam Bourke, Michelle Collinson, Jamie Stokes, Liz Steed, Derek Rosario, Amanda J Farrin, Steph JC Taylor.

**Project administration:** Saïd Ibeggazene, Sophie Reale, Alison Scope, Grace Price, Liam Bourke, Michelle Collinson, Jamie Stokes.

**Writing – original draft:** Saïd Ibeggazene.

**Writing – review & editing:** Sophie Reale, Alison Scope, Grace Price, Eileen Sutton, Liam Bourke, Michelle Collinson, Jamie Stokes, Liz Steed, Derek Rosario, Amanda J Farrin, Steph JC Taylor.

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
