## [Decision Letter · Decision Letter 0]

PONE-D-24-40667Process evaluation for the STAMINA randomised controlled trial: a protocolPLOS ONE

Dear Dr. Ibeggazene,

Thank you for submitting your manuscript to PLOS ONE. After careful consideration, we feel that it has merit but does not fully meet PLOS ONE’s publication criteria as it currently stands. Therefore, we invite you to submit a revised version of the manuscript that addresses the points raised during the review process.

We look forward to receiving your revised manuscript.

Kind regards,

Xing Xiong, M.D.

Academic Editor

PLOS ONE

Journal Requirements:

This work is funded by the NIHR Programme Grants for Applied Research (PGfAR) RP-PG-1016-20007. The views expressed are those of the author(s) and not necessarily those of the NHS, the NIHR or the Department of Health and Social Care.  

3. Ethics statement only appears at the end of the manuscript:

Your ethics statement should only appear in the Methods section of your manuscript. If your ethics statement is written in any section besides the Methods, please move it to the Methods section and delete it from any other section. Please ensure that your ethics statement is included in your manuscript, as the ethics statement entered into the online submission form will not be published alongside your manuscript. 

4. Please include a caption for figure 2.

5. We note you have included a table to which you do not refer in the text of your manuscript. Please ensure that you refer to Table 5, 6, 7 and 9 in your text; if accepted, production will need this reference to link the reader to the Table. 

Reviewers' comments:

Reviewer's Responses to Questions

**Comments to the Author**

1. Does the manuscript provide a valid rationale for the proposed study, with clearly identified and justified research questions?

Reviewer #1: Yes

Reviewer #2: Yes

Reviewer #3: Yes

2. Is the protocol technically sound and planned in a manner that will lead to a meaningful outcome and allow testing the stated hypotheses?

Reviewer #1: Yes

Reviewer #2: Yes

Reviewer #3: Partly

3. Is the methodology feasible and described in sufficient detail to allow the work to be replicable?

Reviewer #1: Yes

Reviewer #2: Yes

Reviewer #3: Yes

4. Have the authors described where all data underlying the findings will be made available when the study is complete?

Reviewer #1: Yes

Reviewer #2: No

Reviewer #3: Yes

5. Is the manuscript presented in an intelligible fashion and written in standard English?

Reviewer #1: Yes

Reviewer #2: Yes

Reviewer #3: Yes

6. Review Comments to the Author

You may also provide optional suggestions and comments to authors that they might find helpful in planning their study.

Reviewer #1: 1.Can the authors clarify the rationale behind the chosen mixed-methods process evaluation design and its alignment with the study's objectives?

2. How do the authors ensure the consistency (fidelity) of training and intervention delivery across different sites, especially since the trial involves multiple healthcare professionals and exercise professionals?

3. The study proposes integrating findings from different methods (e.g., interviews, process outcomes). Can the authors provide more specific details about how they plan to triangulate these findings to draw meaningful conclusions?

4. The authors mention some limitations of their approach, particularly regarding fidelity and the absence of audio recordings of healthcare professional-patient interactions. Could the authors elaborate on alternative strategies to mitigate the effects of these limitations?

Reviewer #2: Well designed evaluation study protocol and well written manuscript. Congratulations. Few comments/suggestions are attached

Page 3-Line no-53-56: just wondering whether the listed adverse events would be evaluated pre and post intervention

Page 7 Line no. 150 and 19 and line number -Implementation team doing the process evaluation-Need to incorporate suggestion to strengthen the manuscript-Are these going to be evaluated pre and post intervention?

Page.7: table numbering to be corrected for column as currently number continues from 1 to 59. Each column and row should start with 1

Page 11 -Line no.163: Omission to be rectified-Add ADT

Page 15 Table 4: Suggestion for inclusion--Facilitating factors that helped/induced sustenance of the participants in the programme could be added

Page 15 Table 4: Repetition to be deleted-Affective attitude and burden

Page 15 Line no. 214 to 215: requires rewording for clarity. Cannot understand the statement made. No. of sites will be reduced to maintain treatment fidelity?

Page 17 Line 266: required additional information. If participant doesnt come for follow up intervention, what steps would be taken-could be described

Reviewer #3: The manuscript outlines a protocol for a process evaluation embedded within a lifestyle intervention trial for prostate cancer patients undergoing androgen deprivation therapy (ADT). This evaluation covers recruitment, fidelity, intervention reach, and organizational aspects. Below are some points for clarification and consideration to enhance the transparency and robustness of the evaluation.

Major Comments:

Validity of the Analysis for Endpoints Based on Questionnaires:

The primary endpoints, quality of life and fatigue, rely on self-reported questionnaires (FACT-Prostate and FACIT-Fatigue), which, while validated, may be subject to response bias and measurement variability. How will the study account for the inherent subjectivity and potential variability in self-reported data? Have any sensitivity analyses been considered to assess the robustness of findings against these potential biases?

Timing and Schedule of Endpoint Collection:

The manuscript mentions that process data will be collected at various points during and after the intervention, but it is not entirely clear how this aligns with the timing of the main trial endpoints (e.g., quality of life measured at 12 months). Could the authors clarify the alignment of process data collection with primary trial outcomes? Additionally, given the 12-month follow-up, how will potential attrition bias be managed, particularly in relation to the consistency of data collected from self-reported questionnaires over time?

Fidelity and Intervention Delivery:

The manuscript highlights fidelity assessment as a key component of the process evaluation, utilizing audio recordings, direct observations, and checklists. Could the authors elaborate on how fidelity will be quantified and reported? Will there be a specific scoring system to assess adherence to intervention components? Additionally, how will deviations from the intervention protocol be addressed within the analysis framework?

Bias in Self-Reported Data:

While the process evaluation is comprehensive, it does not address potential bias in the subjective, self-reported data (e.g., physical activity and quality of life outcomes) and qualitative interviews. How will such bias be accounted for in the study, and what impact might this have on the interpretation of the study’s conclusions?

Minor Comments:

Understanding Trial Recruitment Performance:

The process evaluation likely involves tracking recruitment metrics, assessing participant demographics, and gathering qualitative feedback from recruitment staff. Will the evaluation also examine factors that may hinder recruitment, such as through feedback from recruitment staff and potential participants who declined to participate?

Describing Intervention Reach, Dose Delivered, and Dose Received:

Intervention reach, dose delivered, and dose received will be assessed through metrics such as session attendance and participant engagement. Is there a plan to supplement these metrics with qualitative feedback from participants, such as through exit interviews or surveys, to provide more context on adherence challenges?

Exploring Organizational Implications for Embedding and Sustaining the Intervention:

Semi-structured interviews with HCPs, guided by the Normalization Process Theory (NPT), are intended to assess the feasibility of embedding the intervention within the NHS. Have the authors considered using focus groups alongside individual interviews to capture a broader range of organizational dynamics and collaborative elements?

7. PLOS authors have the option to publish the peer review history of their article (what does this mean? ). If published, this will include your full peer review and any attached files.

**Do you want your identity to be public for this peer review?** For information about this choice, including consent withdrawal, please see our Privacy Policy .

Reviewer #1: No

Reviewer #2: No

Reviewer #3: No

---

## [Author Response · Author response to Decision Letter 1]

6 Dec 2024

Editor:

Thank you for selecting these reviewers. Please find our responses below.

We must raise concerns about the review provided by Reviewer 3. The format and language of the review coupled with the irrelevance of the many of the comments to process evaluations leads us to believe they have used an AI/LLM to generate their response. One possible implication of this is that they have breached confidentiality by providing access to this confidential manuscript to a third party. This would of course be grounds for legal action on the part of the authors. It is disappointing that this has not been picked up by the editorial office.

We are grateful to the other reviewers for their diligent input.

Reviewer #1: Many thanks for taking the time to review our manuscript and providing your input.

1.Can the authors clarify the rationale behind the chosen mixed-methods process evaluation design and its alignment with the study's objectives?

The mixed-methods design was chosen in line with the latest guidance concerning the evaluation of complex interventions and process evaluations. This led us to use the UK Medical Research Council and Linnan and Steckler evaluation frameworks as stated on lines 172-173. It would not be possible to address the various aims of this process evaluation without employing a combination of qualitative and quantitative methods as some aims are incompatible with a particular type of methods. Qualitative methods are often required for questions of understanding. Quantitative methods are well suited to descriptive questions which, given the scale of this evaluation and inevitable resource constraints, offer efficient means of collecting data.

Furthermore, in the seminal paper by Oakley et al 2006 in The BMJ it is clearly articulated that in process evaluations of randomised controlled trials should be informed by both quantitative and qualitative data - https://pmc.ncbi.nlm.nih.gov/articles/PMC1370978/

2. How do the authors ensure the consistency (fidelity) of training and intervention delivery across different sites, especially since the trial involves multiple healthcare professionals and exercise professionals?

The STAMINA trial is a pragmatic study whereby healthcare professionals and exercise professionals (i.e. non-research staff) will deliver the interventions in a real-world situation. As such the potential for variation in the delivery of training and the intervention is to be anticipated. Fidelity is conceptualised in this study according the National Institute of Health Behavioural Change Consortium framework (line 25, line 286 – reference 28) and we sought to preserve fidelity according to the guidance in this framework.

With regards to the fidelity of training – in cases of both healthcare professional training and exercise professional training standardised training packages were used by an intentionally small number of training facilitators using a purpose-designed training manual. In the case of the exercise professional training, all training was delivered by one facilitator. This was not guaranteed by design but will be demonstrated in the published results (see facilitator characteristics in Table 1).

With regards to fidelity of delivery – Healthcare professionals received physical prompts to take to clinic visits to facilitate delivery of the target behaviours in practice. Exercise professionals received a detailed intervention delivery manual with step-by-step instructions regarding the target behaviours. Further details of these are intended to be published in other works.

3. The study proposes integrating findings from different methods (e.g., interviews, process outcomes). Can the authors provide more specific details about how they plan to triangulate these findings to draw meaningful conclusions?

There are several ways by which the design of this evaluation gives us the opportunity to integrate findings and triangulate data from different sources. Firstly, from Table 1 it is clear that there are many instances where data from multiple sources are integrated to provide a more comprehensive insight into the behaviour of individual process evaluation component. Next, the relationship between these components and how they are presumed to act in a causal chain of events that produce the intervention outcomes is described in the programme logic (Fig 1). From this we have a theoretical basis upon which to consider the behaviour of individual components was influenced by others. It is less clear a priori exactly how this will emerge because there are many possible ways that the process data could deviate from a perfectly homogenous prespecified model.

As a simplified single example of how data might be integrated, we may observe that there are high rates of dropout from the intervention that vary by site. Interview data from participants healthcare professionals and/or exercise professionals might elucidate some factors exist may have negative influences on the acceptability of the intervention.

It is probably not worthwhile to try to anticipate all examples of how we might integrate findings because there are too many possibilities. To put it another way there is a both a large potential problem space and solution space meaning that attempting to concisely define all possible problems and potential solutions is impossible.

For the reasons explained above, there are very, very few examples of how process evaluation protocols specify how their results would be integrated. Here is the best example we can find of how this a process evaluation protocol would address your question – clearly there are limitations to the level of detail that can be provided in a such a protocol:

“Integration of the qualitative and quantitative case study data regarding implementation of different components of the intervention will allow detailed evaluation of the quality of intervention implementation in each case study site. For example, quantitative data relating to the number and type of patient problems identified for the agenda and included in the health plan can be integrated with observational data about how clinicians addressed agenda setting and health planning. The additional integration of interview data about patients’ and clinicians’ experiences of reviews will provide in-depth insight into those processes.”

Mann C, Shaw A, Guthrie B, et al. Protocol for a process evaluation of a cluster randomised controlled trial to improve management of multimorbidity in general practice: the 3D study. BMJ Open 2016;6:e011260. doi:10.1136/bmjopen-2016-011260

4. The authors mention some limitations of their approach, particularly regarding fidelity and the absence of audio recordings of healthcare professional-patient interactions. Could the authors elaborate on alternative strategies to mitigate the effects of these limitations?

There is limited evidence-based mitigation possible for mitigated the limitations of this particular method of data collection. We can’t know confidently what the effect the HCPs had on patient behaviours, but we could be explore this in future implementation work if the trial has a positive outcome. Some information regarding professional-patient interactions may be picked up in patient interviews – we explicitly asked patients in interviews about their experiences with HCPs and HCP target behaviours.

Reviewer #2: Many thanks for taking the time to review our manuscript and providing your input.

Well designed evaluation study protocol and well written manuscript. Congratulations. Few comments/suggestions are attached

Page 3-Line no-53-56: just wondering whether the listed adverse events would be evaluated pre and post intervention

Thank you for your kind comment. Only some of these adverse consequences of ADT are evaluated as part of the trial including fatigue, adiposity indices, body mass, sexual dysfunction. Some of these adverse consequences are not suitable to study over only the 12-month follow-up period of this study. Note that the measurement of these outcomes is addressed by the trial protocol and not this process evaluation.

Page 7 Line no. 150 and 19 and line number -Implementation team doing the process evaluation-Need to incorporate suggestion to strengthen the manuscript-Are these going to be evaluated pre and post intervention?

Though it is unclear what is being asked here, please refer to Fig 2 to see the timelines for evaluation of the process evaluation components in relation to intervention delivery.

Page.7: table numbering to be corrected for column as currently number continues from 1 to 59. Each column and row should start with 1

The numbering in Table 1 deliberately continues from 1-59 to facilitate cross referencing of these data sources in Table 2. An alternative numbering of A1, B2, etc could be used instead.

Page 11 -Line no.163: Omission to be rectified-Add ADT

Thank you for spotting this formatting error, it has now been rectified.

Page 15 Table 4: Suggestion for inclusion--Facilitating factors that helped/induced sustenance of the participants in the programme could be added

We appreciate the suggestion to include this factor, but we would like to remain consistent within the externally defined theoretical framework that we are applying – see reference 26 below. In any case topic guides for the interviews have already been approved as part of the ethics application and it may not be timely to request and amendment. The framework was semi-structured so elements related to sustenance might have been captured and can discussed in the study results if appropriate.

Sekhon M, Cartwright M, Francis JJ. Acceptability of healthcare interventions: an overview of reviews and development of a theoretical framework. BMC Health Serv Res. 2017 Jan 26;17(1):88.

Page 15 Table 4: Repetition to be deleted-Affective attitude and burden

Good spot – now removed.

Page 15 Line no. 214 to 215: requires rewording for clarity. Cannot understand the statement made. No. of sites will be reduced to maintain treatment fidelity?

Original statement:

Treatment fidelity is particularly important for multi-site studies to understand if the intervention is operationalised in the same way across sites and minimise site by treatment interactions.

Reworded statement: In multi-site studies, it is particularly important to assess treatment fidelity because if treatment effects vary by site, we can observe whether this is related to variations in the delivery of treatment.

Page 17 Line 266: required additional information. If participant doesnt come for follow up intervention, what steps would be taken-could be described

This question relates more to the trial than the process evaluation and has been addressed in the trial protocol:

“Intervention adherence will be monitored weekly by the research team at SHU by reviewing SLI participants’ attendance at supervised exercise sessions, to identify those who have dipped below 75% attendance. CESs at each NH site will upload data relating to attendance and completion of aerobic and resistance exercise components of the SLI after each session using a REDCap web-based reporting software. Researchers will download, compile, and clean the data on a weekly basis. The database will track adherence relative to the number of prescribed sessions to date and produce “alerts” once a SLI participant drops below 75% adherence. This alert with suggested actions will be sent onto the NH fitness manager who will identify and deliver behavioural support in line with the COM-B [30] model of behaviour change (i.e. data verification, identify required behavioural support and deliver behavioural support). If a participant’s attendance remains below 75%, a maximum two further alerts are sent. A record of all alerts sent and subsequent actions will be maintained.”

McNaught, E., Reale, S., Bourke, L. et al. Supported exercise TrAining for Men wIth prostate caNcer on Androgen deprivation therapy (STAMINA): study protocol for a randomised controlled trial of the clinical and cost-effectiveness of the STAMINA lifestyle intervention compared with optimised usual care, including internal pilot and parallel process evaluation. Trials 25, 257 (2024). https://doi.org/10.1186/s13063-024-07989-y

Reviewer #3: The manuscript outlines a protocol for a process evaluation embedded within a lifestyle intervention trial for prostate cancer patients undergoing androgen deprivation therapy (ADT). This evaluation covers recruitment, fidelity, intervention reach, and organizational aspects. Below are some points for clarification and consideration to enhance the transparency and robustness of the evaluation.

Major Comments:

Validity of the Analysis for Endpoints Based on Questionnaires:

The primary endpoints, quality of life and fatigue, rely on self-reported questionnaires (FACT-Prostate and FACIT-Fatigue), which, while validated, may be subject to response bias and measurement variability. How will the study account for the inherent subjectivity and potential variability in self-reported data? Have any sensitivity analyses been considered to assess the robustness of findings against these potential biases?

This comment is relevant to the design of the trial endpoints but not the objectives of the process evaluation discussed herein.

Timing and Schedule of Endpoint Collection:

The manuscript mentions that process data will be collected at various points during and after the intervention, but it is not entirely clear how this aligns with the timing of the main trial endpoints (e.g., quality of life measured at 12 months). Could the authors clarify the alignment of process data collection with primary trial outcomes? Additionally, given the 12-month follow-up, how will potential attrition bias be managed, particularly in relation to the consistency of data collected from self-reported questionnaires over time?

Again, this comment is relevant to the design of the trial endpoints but not the objectives of the process evaluation discussed herein.

Fidelity and Intervention Delivery:

The manuscript highlights fidelity assessment as a key component of the process evaluation, utilizing audio recordings, direct observations, and checklists. Could the authors elaborate on how fidelity will be quantified and reported? Will there be a specific scoring system to assess adherence to intervention components? Additionally, how will deviations from the intervention protocol be addressed within the analysis framework?

Specific information about how fidelity is being assessed and scored is described in details in the fidelity analysis section – lines 292-296. Deviations from the intervention protocol are a matter for the trial analysis and not the process evaluation and will be described in detail in the statistical analysis plan which is awaiting publication. Where patterns of poor treatment fidelity occur, they will be reported as part of this process evaluation but not necessarily intervened upon.

Bias in Self-Reported Data:

While the process evaluation is comprehensive, it does not address potential bias in the subjective, self-reported data (e.g., physical activity and quality of life outcomes) and qualitative interviews. How will such bias be accounted for in the study, and what impact might this have on the interpretation of the study’s conclusions?

Again, this comment is relevant to the design of the trial endpoints but not the objectives of the process evaluation discussed herein.

Minor Comments:

Understanding Trial Recruitment Performance:

The process evaluation likely involves tracking recruitment metrics, assessing participant demographics, and gathering qualitative feedback from recruitment staff. Will the evaluation also examine factors that may hinder recruitment, such as through feedback from recruitment staff and potential participants who declined to participate?

As stated in Table 1 (58) and Table 3 interview data from healthcare professionals will explore topics such as this. In Table 1 (4) we report that we will collect data about reasons for declining to take part from participants.

Describing Intervention Reach, Dose Delivered, and Dose Received:

Intervention reach, dose delivered, and dose received will be assessed through

---

## [Decision Letter · Decision Letter 1]

PONE-D-24-40667R1Process evaluation for the STAMINA randomised controlled trial: a protocolPLOS ONE

Dear Dr. Ibeggazene,

Thank you for submitting your manuscript to PLOS ONE. After careful consideration, we feel that it has merit but does not fully meet PLOS ONE’s publication criteria as it currently stands. Therefore, we invite you to submit a revised version of the manuscript that addresses the points raised during the review process.

We look forward to receiving your revised manuscript.

Kind regards,

Xing Xiong, M.D.

Academic Editor

PLOS ONE

Journal Requirements:

Reviewers' comments:

Reviewer's Responses to Questions

**Comments to the Author**

1. Does the manuscript provide a valid rationale for the proposed study, with clearly identified and justified research questions?

Reviewer #2: Yes

Reviewer #3: Yes

2. Is the protocol technically sound and planned in a manner that will lead to a meaningful outcome and allow testing the stated hypotheses?

Reviewer #2: Yes

Reviewer #3: Yes

3. Is the methodology feasible and described in sufficient detail to allow the work to be replicable?

Reviewer #2: Yes

Reviewer #3: Yes

4. Have the authors described where all data underlying the findings will be made available when the study is complete?

Reviewer #2: Yes

Reviewer #3: Yes

5. Is the manuscript presented in an intelligible fashion and written in standard English?

Reviewer #2: Yes

Reviewer #3: Yes

6. Review Comments to the Author

You may also provide optional suggestions and comments to authors that they might find helpful in planning their study.

Reviewer #2: It makes sense, from your explanation, to use continuous numbers in table 1, from 1 to 59. So please keep it and not change it to A1, B2, etc which would be more complex.

However, one clarification is requested:

1. "Page 7 Line no. 150 and 19 and line number -Implementation team doing the process evaluation-Need to incorporate suggestion to strengthen the manuscript-Are these going to be evaluated pre and post intervention?

Though it is unclear what is being asked here, please refer to Fig 2 to see the timelines for evaluation of the process evaluation components in relation to intervention delivery."

To clarify my query-This is regarding using implementation team for process evaluation. Though this is not ideal, funding limitation is an understandable reason for this compromise, which has already been mentioned as a limitation. What my suggestion regarding this is to add few precautions/strategies that would be undertaken to reduce bias in process evaluation reporting.

Reviewer #3: Thank you for addressing all the comments.

The responses adequately address the reviewer’s concerns and provide relevant updates to the manuscript. The authors demonstrate an awareness of the study’s exploratory nature and the inherent limitations, which is appropriate for the study design.

7. PLOS authors have the option to publish the peer review history of their article (what does this mean? ). If published, this will include your full peer review and any attached files.

**Do you want your identity to be public for this peer review?** For information about this choice, including consent withdrawal, please see our Privacy Policy .

Reviewer #2: No

Reviewer #3: No

---

## [Author Response · Author response to Decision Letter 2]

20 Mar 2025

Reviewer #2: It makes sense, from your explanation, to use continuous numbers in table 1, from 1 to 59. So please keep it and not change it to A1, B2, etc which would be more complex.

However, one clarification is requested:

1. "Page 7 Line no. 150 and 19 and line number -Implementation team doing the process evaluation-Need to incorporate suggestion to strengthen the manuscript-Are these going to be evaluated pre and post intervention?

Though it is unclear what is being asked here, please refer to Fig 2 to see the timelines for evaluation of the process evaluation components in relation to intervention delivery."

To clarify my query-This is regarding using implementation team for process evaluation. Though this is not ideal, funding limitation is an understandable reason for this compromise, which has already been mentioned as a limitation. What my suggestion regarding this is to add few precautions/strategies that would be undertaken to reduce bias in process evaluation reporting.

Thank you for the clarification. We believe the main strategy that we are able to employ to reduce reporting biases is to publish this protocol as now described on Pg 27, L313-317.

“….which has the potential to create biases in the collection and reporting of process evaluation data. In publishing this protocol, we intend to mitigate against the impact of these potential biases by acknowledging their potential in the data collection process for and by making explicit what data is to be collected, how each data type will be used to describe specific process evaluation components and how the process evaluation components and frameworks relate to each other.”

Publishing process evaluation protocols is still a rare practice with no guidelines though it is becoming more common.

Other methods to reduce reporting biases include inclusion of the interview topic guides in the Supplementary materials. We have also already stated that “At least 20% of data will be independently double or triple coded” with regard to interview analysis. We also have made clear where our findings will be triangulated using data from multiple sources (table 1).

Reviewer #3: Thank you for addressing all the comments.

The responses adequately address the reviewer’s concerns and provide relevant updates to the manuscript. The authors demonstrate an awareness of the study’s exploratory nature and the inherent limitations, which is appropriate for the study design.

---

## [Decision Letter · Decision Letter 2]

Process evaluation for the STAMINA randomised controlled trial: a protocol

PONE-D-24-40667R2

Dear Dr. Ibeggazene,

We’re pleased to inform you that your manuscript has been judged scientifically suitable for publication and will be formally accepted for publication once it meets all outstanding technical requirements.

Kind regards,

Xing-Xiong An, M.D.

Academic Editor

PLOS ONE

Additional Editor Comments (optional):

Your paper has now been approved by the reviewer, and I am pleased to inform you that your manuscript can be accepted for publication now.

Reviewers' comments:

Reviewer's Responses to Questions

**Comments to the Author**

1. Does the manuscript provide a valid rationale for the proposed study, with clearly identified and justified research questions?

Reviewer #2: Yes

2. Is the protocol technically sound and planned in a manner that will lead to a meaningful outcome and allow testing the stated hypotheses?

Reviewer #2: Yes

3. Is the methodology feasible and described in sufficient detail to allow the work to be replicable?

Reviewer #2: Yes

4. Have the authors described where all data underlying the findings will be made available when the study is complete?

Reviewer #2: No

5. Is the manuscript presented in an intelligible fashion and written in standard English?

Reviewer #2: Yes

6. Review Comments to the Author

You may also provide optional suggestions and comments to authors that they might find helpful in planning their study.

Reviewer #2: Thank you for revising the manuscript to address the queries and suggestions. I donot have any more queries.

7. PLOS authors have the option to publish the peer review history of their article (what does this mean? ). If published, this will include your full peer review and any attached files.

**Do you want your identity to be public for this peer review?** For information about this choice, including consent withdrawal, please see our Privacy Policy .

Reviewer #2: No

---

## [Editor Report · Acceptance letter]

PONE-D-24-40667R2

PLOS ONE

Dear Dr. Ibeggazene,

I'm pleased to inform you that your manuscript has been deemed suitable for publication in PLOS ONE. Congratulations! Your manuscript is now being handed over to our production team.

Kind regards,

on behalf of

Dr. Xing-Xiong An

Academic Editor

PLOS ONE